# What is a minimal clinically important difference for clinical trials in patients with disorders of consciousness? a novel probabilistic approach

**Martin M. Monti**[1,2]*, **Norman M. Spivak**[2,3], **Brian L. Edlow**[4,5], **Yelena G. Bodien**[4,6]

1 Department of Psychology, University of California Los Angeles, Los Angeles, California, United States of America, 2 Department of Neurosurgery, Brain Injury Research Center, University of California Los Angeles, Los Angeles, California, United States of America, 3 UCLA-Caltech Medical Scientist Training Program, David Geffen School of Medicine, UCLA, Los Angeles, California, United States of America, 4 Department of Neurology, Center for Neurotechnology and Neurorecovery, Massachusetts General Hospital and Harvard Medical School, Boston, Massachusetts, United States of America, 5 Athinoula A. Martinos Center for Biomedical Imaging, Massachusetts General Hospital, Charlestown, Massachusetts, United States of America, 6 Department of Physical Medicine and Rehabilitation, Spaulding Rehabilitation Hospital and Harvard Medical School Charlestown, Massachusetts, United States of America

* monti@ucla.edu

**Data Availability Statement:** The code for calculating pMCID (pMCID_inDOC.m), including the code necessary to reproduce the results shown in Tables 2, 3, and 4, is freely available from the

## Abstract

Over the last 30 years, there has been a growing trend in clinical trials towards assessing novel interventions not only against the benchmark of statistical significance, but also with respect to whether they lead to clinically meaningful changes for patients. In the context of Disorders of Consciousness (DOC), despite a growing landscape of experimental interventions, there is no agreed standard as to what counts as a minimal clinically important difference (MCID). In part, this issue springs from the fact that, by definition, DOC patients are either unresponsive (i.e., in a Vegetative State; VS) or non-communicative (i.e., in a Minimally Conscious State; MCS), which renders it impossible to assess any subjective perception of benefit, one of the two core aspects of MCIDs. Here, we develop a novel approach that leverages published, international diagnostic guidelines to establish a probability-based minimal clinically important difference (pMCID), and we apply it to the most validated and frequently used scale in DOC: the Coma Recovery Scale–Revised (CRS-R). This novel method is objective (i.e., based on published criteria for patient diagnosis) and easy to recalculate as the field refines its agreed-upon criteria for diagnosis. We believe this new approach can help clinicians determine whether observed changes in patients' behavior are clinically important, even when patients cannot communicate their experiences, and can align the landscape of clinical trials in DOC with the practices in other medical fields.

corresponding author (MMM) and at the following OSF link: https://doi.org/10.17605/OSF.IO/D2FYM.

**Funding:** MMM Funding: Tiny Blue Dot Foundation, NIH National Institute of General Medical Sciences (5R01GM135420). NMS Funding: NIH National Institute of General Medical Sciences (5T32GM008042). BLE Funding: NIH Director's Office (DP2HD101400), NIH National Institute of Neurological Disorders and Stroke (R21NS109627, RF1NS115268), James S. McDonnell Foundation, Tiny Blue Dot Foundation, and Chen Institute MGH Research Scholar Award. YB Funding: NIDILRR 90DPTB0011-01-00, Department of Defense (W81XWH-14-2-0176), NIH 1DP2HD101400, Tiny Blue Dot Foundation, James S McDonnell Foundation.

**Competing interests:** The authors have declared that no competing interests exist.

## Introduction

In a landscape of ever-increasing sophistication in medical technology, the design of ethical clinical trials requires balancing the risks and benefits of testing novel interventions [1]. Historically, novel investigational interventions are evaluated against the benchmark of statistical significance. That is, they are typically considered successful when an observed difference, across cohorts (e.g., intervention group versus placebo group) or over time (e.g., prior to intervention versus following intervention), is sufficiently large compared to its variability. While this approach is sound with respect to well-established criteria for making inferences based on random samples [2], it is blind as to whether the observed changes are meaningful for patients, families, clinicians, decision-makers, and other stakeholders [3,4]. Over the last thirty years there has been a growing emphasis on evaluating the clinical meaningfulness of the changes associated with investigational interventions alongside, or in lieu of, statistical significance. Indeed, thresholds establishing the minimal effect magnitude associated with a meaningful change (i.e., Minimal Clinically Important Difference; MCID) [5] have now been established for clinical instruments across a broad range of outcomes, including level of patient disability (e..g, Functional Independence Measure [6]), balance (e.g., Berg Balance Scale [7]), depression (e.g., Montgomery–Asberg Depression Rating Scale [8]), motor deficits (e.g., Disability Rating Scale [9,10]) and quality of life (e.g., Heinrichs–Carpenter quality of life scale [11]), among many others.

For patients with disorders of consciousness (DOC) [12,13], despite a burgeoning literature assessing novel therapeutic interventions [1,14,15], there are no accepted MCIDs for the most highly recommended and frequently used instrument [16,17]: the Coma Recovery Scale–Revised (CRS-R) [18]. To date, only one DOC-specific instrument has a published MCID, the Disorders of Consciousness Scale (DOCS-25; [19]). However, a comprehensive review by the American Congress of Rehabilitation Medicine (ACRM) found the CRS-R to be the preferred scale to assess patients with DOC [20]. Consequently, clinical trials in this patient cohort have relied exclusively on statistical significance when assessing the efficacy of pharmacological [21,22], device-based [23–25], and sensory stimulation interventions [26,27].

Establishing MCIDs in patients with DOC is hampered by two main difficulties. First, one of the two core aspects of MCIDs is not accessible in this patient population. Specifically, as originally defined, an MCID is "[t]he smallest difference in score in the domain of interest *which patients perceive as beneficial* and which would mandate, in the absence of troublesome side effects and excessive cost, a change in the patient's management" [emphasis added] [5]. By the very nature of the diagnosis, patients with DOC are either unresponsive or responsive but non-communicative [28]. *Ipso facto*, subjective perceptions of improvement by the patient are either absent (in unconscious patients) or inaccessible (in non-communicative patients). Furthermore, patients who emerge from the minimally conscious state (MCS) and recover basic functional communication often remain in a confusional state, characterized by disorientation and severe impairments in cogntion [29–31], all known limitations in the development of reliable MCIDs [32]. The second core aspect of an MCID, is "[t]he smallest difference [. . .] which would mandate, in the absence of troublesome side effects and excessive cost, *a change in the patient's management*," [5] [emphasis added] and can, in principle, be evaluated in non-responsive, non-communicative, and severely disabled patients. Specifically, given the existence of well-defined behaviors that signal a patient's DOC diagnosis [28], it is possible to operationalize 'clinically meaningful changes' in this context as any change–following an intervention–whereby a patient becomes capable of demonstrating novel behaviors that are associated with attaining a diagnostic status implying lesser impairment (i.e., henceforth, a "threshold behavior"). To exemplify, according to the CRS-R protocol [18], when a patient

considered to be in a Vegetative State (VS; also referred to as Unresponsive Wakefulness Syndrome; UWS [33]) regains the ability to demonstrate visual fixation, this signals a change in diagnostic status from VS to Minimally Conscious State minus (MCS-). Similarly, when a patient considered to be in a Minimally Conscious State plus (MCS+) regains the ability to demonstrate accurate functional communication or functional object use, this signals a change in diagnostic status from MCS+ to emergence from a Minimally Conscious State (eMCS) [18,28,34,35]. In this proposal, the emergence of such "threshold behaviors," which imply a change in diagnosis and thus patient management, can be leveraged to identify changes in CRS-R total scores that are clinically meaningful.

The second main difficulty in developing MCIDs for DOC patients is that clinical instruments for this cohort typically describe patient responsiveness in terms of ordinally ranked behaviors, from highest to lowest. The distance between contiguous items on the scale, however, is not designed to be proportional and/or consistent. This characteristic makes it difficult to develop a simple numeric threshold that can be used to unambiguously define clinically meaningful changes across the entirety of the instrument's diagnostic range. To illustrate, a 1-point improvement at the upper end of the Oromotor/Verbal Function subscale of the CRS-R, from 2 (i.e., Vocalization/Oral Movement) to 3 (i.e., Intelligible Verbalization), seems very different from a 1-point improvement on the lower end of the same subscale from 0 (i.e., No response) to 1 (i.e., Oral Reflexive Movement). For, while the former change implies the return of language comprehension and a state of awareness, also denoting a change in diagnosis from VS to MCS+, the latter scenario simply represents a change from complete unresponsiveness (in this functional domain) to simple reflexive responsiveness. In fact, while some 1-point increases in CRS-R score score can entail a clinically important change, as in the former scenario, there are many instances in which a 1- and even 2-point increase is not associated with any clinically meaningful change. It is thus difficult to find a single minimal change threshold that is reliably associated with a clinically meaningful change.

Given the above, it is difficult to deploy existing procedures for developing MCIDs [36–38] in DOC patients. The distribution-based and the anchor methods, for example, require input from patients regarding their baseline state and any noticeable improvement they might perceive, which, as discussed above, is either absent or inaccessible in DOC patients. The Delphi method is feasible in this context, but relies on subjective clinical opinion and is susceptible to individual biases and that are not unusual in the context of DOC even among experts [39–42]. In what follows, we thus developed a novel, clinical guideline-driven [12], probabilistic approach to establishing MCIDs (pMCID) in DOC. Specifically, we leverage the 2002 Aspen Neurobehavioral Conference Workgroup criteria [28] to identify unambiguous "threshold behaviors" which, when attained, imply a change in patient diagnosis and thus a change in clinical management (i.e., a clinically important change), and then calculate the probability that observing an increase of 1-, 2-, or 3-point change in the total score of the CRS-R is associated with the patient attaining a novel threshold behavior.

## Materials and methods

We use a 2-step procedure to developed a theoretically-driven probabilistic approach to establishing an MCIDs for the CRS-R.

### Selection of threshold behaviors

As shown in Table 1, of the 29 behaviors that can be scored on the CRS-R, eight represent threshold behaviors. Namely, attaining Localization to Noxious Stimulation or Visual Fixation, index a progression from VS to MCS-; Reproducible Movement to Command, Object

**Table 1. List of behaviors tested on the CRS-R protocol and associated scores.** Threshold behaviors and associated diagnosis are shown in bold.

| AUDITORY FUNCTION SCALE | | Associated diagnosis |
|---|---|---|
| 4 | Consistent Movement to Command | |
| **3** | **Reproducible Movement to Command** | **MCS+** |
| 2 | Localization to Sound | |
| 1 | Auditory Startle | |
| 0 | None | |
| VISUAL FUNCTION SCALE | | |
| **5** | **Object Recognition** | **MCS+** |
| 4 | Object Localization: Reaching | |
| 3 | Visual Pursuit | |
| **2** | **Fixation** | **MCS-** |
| 1 | Visual Startle | |
| 0 | None | |
| MOTOR FUNCTION SCALE | | |
| **6** | **Functional Object Use** | **eMCS** |
| 5 | Automatic Motor Response | |
| 4 | Object Manipulation | |
| **3** | **Localization to Noxious Stimulation** | **MCS-** |
| 2 | Flexion Withdrawal | |
| 1 | Abnormal Posturing | |
| 0 | None | |
| OROMOTOR/VERBAL FUNCTION SCALE | | |
| **3** | **Intelligible Verbalization** | **MCS+** |
| 2 | Vocalization/Oral Movement | |
| 1 | Oral Reflexive Movement | |
| 0 | None | |
| COMMUNICATION SCALE | | |
| **2** | **Functional: Accurate** | **eMCS** |
| **1** | **Non-functional: Intentional** | **MCS+** |
| 0 | None | |
| AROUSAL SCALE | | |
| 3 | Attention | |
| 2 | Eye Opening w/o Stimulation | |
| 1 | Eye Opening with Stimulation | |
| 0 | Unarousable | |

Recognition, Intelligible Verbalization, and Intentional (non-functional) communication, index progression to MCS+; and demonstrating Functional Object Use or Accurate (functional) Communication, indexes emergence from a DOC (i.e., eMCS). Threshold behaviors were selected strictly on the basis of accepted published criteria for behviors diagnostic of each DOC (i.e., VS, MCS-, MCS+) [12,13,28,34,43,44]. It should be noted that there are additional behaviors, in the CRS-R, that are consistent with each of the diagnosis above. For example, with respect to the Motor Function Scale, Object Manipulation and Automatic Motor Response are also consistent with an MCS- diagnosis. However, once a patient is capable of Localization to Noxious Stimulation, also demonstrating either of these two additional behaviors does not affect their diagnostic status. In this respect, we only consider threshold behaviors the lowest behavior on each scale that warrants a change in diagnosis.

## Calculation of a probabilistic MCID (pMCID)

After having selected the threshold behaviors on the basis of published guidelines, we use simple combinatorics to calculate the probability that any 1-, 2-, and 3-point change in the CRS-R total score is accompanied by the patient exhibiting at least one novel threshold behavior warranting a change in diagnosis. To achieve this, we start by calculating all the combinations of scores that can exist given the structure of the CRS-R scale (i.e., 29 items subdivided into 6 functional domains). In other words, we start by creating a catalog of all conceivable CRS-R configurations. Then, for each of these configurations, we count the fraction of 1-, 2-, and 3-point changes that are associated with meeting at least one new threshold behavior, as a proportion of all the possible 1-, 2-, and 3-point changes that can be attained form that configuration. To exemplify, given an initial CRS-R total score of 6, obtained by scoring 2, 1, 1, 1, 0, 1 on the auditory, visual, motor, oromotor/verbal, communication, and arousal scales respectively, there are six possible ways in which the total score can increase by 1 point (i.e., one per scale). Of these only three are associated with attaining a new threshold behavior (i.e., a 1-point change in the auditory, visual, and communication scale). Thus, given the above initial CRS-R configuration, a 1-point change is associated 50% of the times with a clinically important change. See also Fig 1 for a pictorial illustration of the calculation as applied to 1-point changes. We repeat this process for each conceivable CRS-R configuration and then calculate the overall probability that a 1-, 2-, and 3-point change is associated, for each diagnostic group

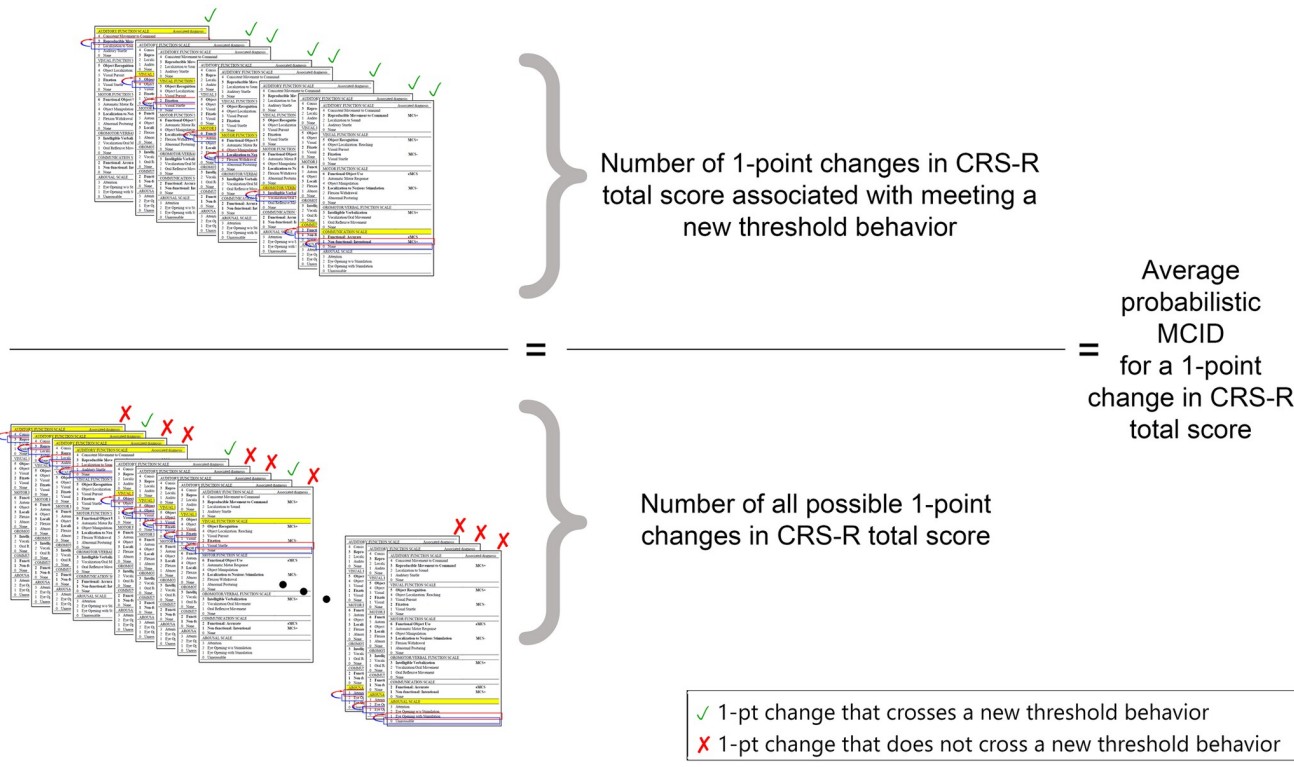

**Fig 1. pMCID depiction.** Sample calculation of the average probabilistic minimal clinically important difference (pMCID) for a 1-point change in CRS-R total score across all diagnoses. In brief, the pMCID for a 1-point change is calculated by taking the ratio of the number of 1-point changes that result in a new threshold behavior (numerator) over the number of all possible 1-point changes (denominator). (Note: While not depicted, the calculation also includes "net" 1-point changes; e.g., where a 1-point decrease is observed in one subscale and a 2-point increase is observed on a different subscale; see methods for full detail).

separately, with an important clinical change. The calculations embed the following four, conservative, stipulations (see the discussion section for an analysis of the impact of these restrictions on the pMCID). First, we do not place any constraint on how the 1-, 2-, and 3-point changes from each initial CRS-R is achieved (e.g., a 3-point change can be obtained by a 3-point increase on a single scale, a 2-point increase in one scale and a 1-point increase on another scale, or a 1-point increase on 3 different scales). Second, we only calculate the probability that each change is associated with at least one threshold behavior. That is, 2- and 3-point changes associated with attaining multiple novel threshold behaviors are treated equally to changes associated with attaining only one novel threshold behavior. Third, changes associated with attaining novel threshold behaviors for the diagnostic group a patient is already part of, or for a lower diagnostic group, are not considered to be clinically important. That is, if a patient is considerd to be MCS+, changes associated with attaining additional threshold behaviors consistent with this diagnosis, as well as changes associated with attaining threshold behaviors consistent with lower diagnoses (e.g., MCS-), are not considered to be clinically important. Finally, our calculations are based on net changes is CRS-R total score. We thus admit, as often seen in the clinic, for scores on individual scales to vary, by up to 3 points, in either direction. That is, a 2-point net change could result by a 3-point increase across two scales and a 1-point decrease on a third scale.

The purely combinatorial approach described above, however, whereby we start by computing all existing configurations of scores across the 6 CRS-R scales, can give rise to impossible score combinations; for example, a CRS-R total score of 6 obtained by attaining Functional Object Use in the Motor Function Scale and 0 on all other scales [45]. We thus repeat the calculation of the proportion of changes associated with attaining novel threshold behaviors for "all existing combinations" as well as for "possible combinations only" (as defined in Ref. [45]). That means, that while in the "all existing combinations" we calculate the pMCID without regard for whether the combination of CRS-R subscores are clinically possible, in the "possible combinations only", we omit from the pMCID calculus any clinical combination of CRS-R subscores that is manifestly impossible. In addition, as an example of the flexibility and "future compatibility" of the present probabilistic approach, we show how easy it is to recalculate these proportions should diagnostic criteria for DOC change. Specifically, it has been recently proposed that the item "Consistent Movement to Command," in the auditory function scale of the CRS-R (cf., Table 1), should be considered to be indicative of emergence from MCS (i.e., a threshold behavior for eMCS) as opposed to MCS+ [46,47]. We thus re-calculate the proportion of changes associated with attaining novel threshold behaviors under this new potential guideline (under the "possible combinations only" regime).

## Results

When considering all existing combinations, there are 10,080 possible CRS-R configurations. As shown in Table 2, a 1-point change in total CRS-R score is associated with a clinically

**Table 2. Probability of patient attaining a new threshold behavior given an observed 1-, 2-, and 3-point change in CRS-R (including all combinations).**

| Diagnosis | Observed change in CRS-R total score | | |
|---|---|---|---|
| | **1 point** | **2 points** | **3 points** |
| VS | 81.39 | 88.84 | 93.77 |
| MCS- | 84.93 | 88.72 | 91.89 |
| MCS+ | 34.15 | 40.40 | 46.75 |
| *Average* | *66.82* | *72.66* | *77.47* |

**Table 3. Probability of patient attaining a new threshold behavior given an observed 1-, 2-, and 3-point change in CRS-R (excluding impossible combinations).**

| Diagnosis | Observed change in CRS-R total score | | |
|---|---|---|---|
| | 1 point | 2 points | 3 points |
| VS | 72.78 | 82.36 | 89.56 |
| MCS- | 74.86 | 80.43 | 85.47 |
| MCS+ | 32.73 | 39.48 | 46.34 |
| *Average* | *60.12* | *67.42* | *73.79* |

important change in 81% and 85% of the times, for VS and MCS- patients respectively, and only in 34% of the times for MCS+ patients. In VS and MCS- patients, 2- and 3-point changes are associated with 89% and over 91% chance of a clinically important change, while, in MCS + patients, 2- and 3-point changes are only associated with a 40% and 47% chance of a clinically important change, respectively.

Out of the 10,080 existing CRS-R configurations, however, according to published data [45], 4,368 are clinically impossible, thus leaving only 5,712 relevant CRS-R configurations. As shown in Table 3, when only considering possible CRS-R configurations, a 1-point change in total score is associated with a clinically important change in 72% and 75% of the times, for VS and MCS- patients respectively, and in 33% of the times for MCS+ patients. In VS and MCS- patients, 2- and 3-point changes are associated with over 80% and 85% chance of a clinically important change, whereas in MCS+ patients 2- and 3-point changes are only associated with a 39% and 46% chance of a clinically important change.

Finally, as shown in Table 4, should international guidelines recognize "Consistent Movement to Command" as a threshold behavior for eMCS [46,47], it would not affect the probabilities reported above for VS and MCS- patients, but it would increase the likelihood of a 1-, 2-, and 3-point change in CRS-R total score being associated with a clinically important change in MCS+ patients to 69%, 72%, and 75% (from 33%, 49%, and 56%), respectively.

## Discussion

Given the difficulty in applying established procedures for developing MCIDs [36–38] in DOC patients, we develop a novel probabilistic MCIDs (pMCID) which can be deployed in non-responsive and non-communicative patients and on an ordinal instrument such as the CRS-R. We do so by leveraging well-established diagnostic criteria distinguishing levels of impairment of consciousness [12] in order to calculate the likelihood that any 1-, 2-, or 3-point increase in the CRS-R total score is associated with a clinically meaningful change. Our results show that, under the current guidelines, it might not be appropriate to use a single MCID threshold

**Table 4. Probability of patient attaining a new threshold behavior given an observed 1-, 2-, and 3-point change in CRS-R (excluding impossible combinations) if Consistent Movement to Command is reassessed to index eMCS.**

| Diagnosis | Observed change in CRS-R total score | | |
|---|---|---|---|
| | 1 point | 2 points | 3 points |
| VS | 72.78 | 82.36 | 89.56 |
| MCS- | 74.86 | 80.43 | 85.47 |
| MCS+ | 69.07 | 71.64 | 74.78 |
| *Average* | *72.23* | *78.14* | *83.27* |

across the diagnostic spectrum covered by the CRS-R protocol. For example, while a 2-point change in the CRS-R total score is already associated with an over 80% chance of attaining a new threshold behavior for VS and MCS- patients, it is only associated with a 40% chance in MCS+ patients. It is thus not obvious how, in this circumstance, a single MCID threshold could be adopted. The reason for this is that out of the 29 CRS-R items, currently only two are threshold behaviors for eMCS, as compared to 5 and 6 for MCS+ and MCS-, respectively. Second, the data presented here suggest that at least for VS and MCS- a 2-point change is sufficient to return an 80% chance of attaining a clinically meaningful new threshold behavior, which could be a reasonable pMCID. For MCS+ patients, however, current guidelines do not provide a threshold yielding a comparable likeliness of attaining a clinically meaningful change. Nonetheless, should diagnostic guidelines be revised to reclassify Consistent Movement to Command as a threshold behavior for eMCS, as previouslyadvocated [46,47], a 2-to-3 point change in total CRS-R would be associated with an over 70% chance of a clinically meaningful change. Furthermore, under this latter scenario, a 2-point change in CRS-R total score would also happen to be associated–across all three diagnostic groups–with a 78% chance of a clinically meaningful change, which could provide an acceptable, pragmatically simpler, pMCID (i.e., applicable regardless of a patient's initial diagnosis).

The flexibility demonstrated by this last example is perhaps the most important feature of the method we are presenting. While it has now been over 20 years from the initial distinction of MCS from VS [28], the nosology of DOC has been refined over time [34] and the relationship between individual items on the CRS-R scale (e.g., visual fixation, localization to noxious stimulation, consistent response to command) and diagnosis is still being discussed (cf., [46,47]) and could well change as more evidence and novel techniques are introduced. Furthermore, while we currently focus exclusively on diagnostic behaviors as markers of clinically important changes, our method could be easily extended to include the views of other stakeholders, such as family members, on which items they would consider particularly valuable.

In evaluating these results, a number of important considerations need to be made with respect to the effect of the stipulations we described in the methods section on the resulting pMCID. Specifically, we highlight that our approach is intrinsically very conservative and thus likely to be an overestimate of the change necessary in CRS-R total score to achieve an MCID in DOC patients. First, our approach recognizes only diagnosis-changing behaviors as meeting the threshold for being "clinically meaningful." It is possible, however, that attaining some non-diagnosis altering behaviors can have meaningful implications. For example, for a patient considered to be MCS+ on the basis of object recognition, the emergence of Intentional (albeit non-functional) Communication or Intelligible Verbalization, is valued by families [48] and can have implications for access to speech and language therapy. Future work combining our quantitative approach with a large sampling of subjective judgments from all stakeholders (e.g., clinicians, care-givers, family members, recovered patients), may lead to a less stringent threshold for the definition of what ought to count as a clinically meaningful change.

Second, the pMCID calculation does not differentiate cases in which one new threshold-behavior is met as compared to cases where multiple new threshold-behaviors are met by a patient. Yet, intuitively, the emergence of multiple novel threshold behaviors, something which is often observed [49], may be associated with better access to multiple pathways for intervention and better prognosis.

Third, we did not count recovery of a threshold behavior as being clinically important if there was no change in diagnosis. For example, should a patient be diagnosed as MCS- on the basis of their ability to demonstrate Localization to Noxious Stimulation, we did not consider the emergence of Object Manipulation or Automatic Motor Responses, behaviors also consistent with MCS-, as being "clinically meaningful" in our approach. Yet, both are novel

behaviors that implicate greater cognitive function and that can be leveraged in the context of rehabilitation. Our conservative choice suggests that the pMCIDs may underestimate, but would not overestimate a clinically meaningful change.

In addition, we also note that while the pMCID developed above relies on the observation of motor-response-dependent (responsive) behaviors. Yet, a growing literature demonstrates that it is possible for DOC patients to be unresponsive in traditional bedside assessments while demonstrating responsiveness through neuroimaging and electrophysiology-based approaches [50] as well as enhanced technology-based monitoring (e.g., video eye-tracking) [51]. In as much as it is possible to map such covert responses to items of the CRS-R (e.g., eye-tracking, response to command), the approach developed above should generalize to patients with covert awareness or cognitive-motor dissociation [52]. The degree to which other neuroimaging or neurophysiology-based biomarkers (e.g., resting state network connectivity) which might carry diagnostic or prognostic value [49], but do not map onto any item of existing neurobehavioral scales, could be incorporated in the above framework remains to be understood.

It should also be considered that, as conceived presently, our approach is valid only inasmuch as the CRS-R is applied appropriately. Specifically, the pMCID has no embedded mechanism to account for confounding factors such as those typical of the acute care setting (e.g., sedation, intracranial pressure, hemodynamics, etc). In this respect, our approach relies on being able to abide by well-recognized best practices for CRS-R assessments [12], which are difficult to meet in acute care, potentially limiting the appropriateness of the pMCID to clinical trials in the subacute and chronic phases of the disease. In partial mitigation of this concern, the CRS-R now contains completion codes aimed at identifying (and removing) potential confound sources [53]. In addition, our approach should also be easy to translate to shortened protocols directly derived from the CRS-R, such as the CRSR-FAST [54] and the SECONDs [55].

Finally, by virtue of calculating the pMCID on the basis of relatively large "all-or-none" diagnostic categories, this approach fails to capture finer changes in level of consciousness and what is likely to be a more continuous spectrum of impairments [56]. However, we highlight that this limitation is due to the choice of using well-established diagnostic markers as the basis for threshold behaviors. As the definition of what should count as a change that "would mandate [. . .] a change in the patient's management" [5] becomes more sophisticated, the pMCID method can be adapted to reflect such finer gradations.

## Conclusion

This is the first paper reporting a CRS-R MCID for evaluating treatment efficacy in patients with DOC. Because patients with DOC are inherently unable to provide reliable responses to questions about meaningful recovery, we derived a pMCID based on the liklihood of patients attaining target behaviors that lead to a change in clinical management. The CRS-R pMCIDs proposed here may be used to define intervention responsiveness in future clinical trials and in investigational drug and device applications to the U.S. Food and Drug Administration. As per Table 3, for example, a 2-point change in total CRS-R score is 82% likely to be associated with a clinically important change in VS patients. Thus, if an intervention gave rise to a 2-point change across a sample of patients, it would have to be considered successful. In addition, the tool could also be modified to allow research teams to create pMCIDs tailored to their research question (i.e., by specifying threshold behaviors that are relevant to their specific research question).

## Author Contributions

**Conceptualization:** Martin M. Monti, Norman M. Spivak, Brian L. Edlow, Yelena G. Bodien.

**Formal analysis:** Martin M. Monti.

**Funding acquisition:** Martin M. Monti, Norman M. Spivak, Brian L. Edlow, Yelena G. Bodien.

**Methodology:** Martin M. Monti, Norman M. Spivak, Brian L. Edlow, Yelena G. Bodien.

**Software:** Martin M. Monti.

**Supervision:** Martin M. Monti, Yelena G. Bodien.

**Validation:** Martin M. Monti.

**Writing – original draft:** Martin M. Monti.

**Writing – review & editing:** Martin M. Monti, Norman M. Spivak, Brian L. Edlow, Yelena G. Bodien.

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
