## [Decision Letter · Decision Letter 0]

22 Jun 2023

PONE-D-23-10843What is a minimal clinically important difference for clinical trials in patients with Disorders of Consciousness? A novel probabilistic approachPLOS ONE

Dear Dr. Monti,

Thank you for submitting your manuscript to PLOS ONE. After careful consideration, we feel that it has merit but does not fully meet PLOS ONE’s publication criteria as it currently stands. Therefore, we invite you to submit a revised version of the manuscript that addresses the points raised during the review process.

We look forward to receiving your revised manuscript.

Kind regards,

Soojin Park, M.D.

Academic Editor

PLOS ONE

Additional Editor Comments:

Reviewer 1:

Monti et al. present a paper in which they develop a probabilistic score of the minimal clinically important difference (MCID) after a therapeutic intervention in patients with disorders of consciousness (DoC). They elegantly introduce the intrinsic challenges of identifying this threshold (i.e., DoC patients are either unaware or low-responsive, which means they are unable to provide subjective reports of perceived benefit), and they continue to solve this problem in a most convincing way by benchmarking their probabilistic MCID score against the clinical gold standard scale of DoC, the Consciousness Recovery Scale revised (CRS-R).

The paper is timely, important, and well-written. I am convinced that it will become an invaluable and well-cited resource for therapeutic DoC trials in the future.

I have only some very minor suggestions:

1. According to the CRS-R, visual fixation is enough to diagnose MCS- but other sources have argued that visual fixation might be a reflexive behavior and that visual pursuit is required for an MCS- diagnosis. What is the authors’ opinion?

2. Being a reviewer from Europe, I am used to the term “VS/UWS” instead of “VS” given the latter’s negative connotations. “VS” is the established term in the US, but for the sake of international outreach, the authors might want to consider changing the terminology. (I know, I know… “VS/UWS” sounds awkward, but still.)

3. The differentiation between VS/UWS and MCS is most probably gradual (continuous) rather than binary (all-or none), and the level of arousal and responsiveness in DoC patients can be subject to fluctuations over very short time spans (minutes, hours), which the MCID score for obvious reasons cannot take into account. Perhaps the authors might want to include a statement about this limitation.

4. Conclusion, line 307: “This is the first study reporting…” – I am not sure I would call this a “study” since the authors did not collect original data or tested their score in a patient cohort. I would suggest changing the sentence to “We report…” or “This is the first paper reporting…”

5. Lines 129/1300: Small typo, should be ”In what follows, we..”

6. The authors provide a link to OSF with their code to calculate the MCID score, which is excellent. A graphical depiction of the idea/concept/design of the MCID score and its calculation would be great. It's optional, but the visual appeal of the paper could be increased a lot with a nice illustration.

Reviewer 2:

I read Dr. Monti et al manuscript with interest. This is an extremely important topic, and I am grateful for the authors addressing and proposing their approach of pMCID in DOC patients. I congratulate them on this very well written paper.

The two challenges as explained elegantly by the authors in the MCID concept in DOC patients are: 1/ the subjective perception of improvement by the patient is absent or not accessible, 2/ the CRS-R is not designed to be proportional and/or consistent. Thus, the authors have developed a probabilistic approach to establishing MCID (pMCID) for the CRS-R. The authors selected 8 threshold behaviors that reflect the behavioral diagnostic of each DOC (VS, MCS-, and MCS+). Tables 3 &4 reflect the probability of patient attaining a new threshold behavior given an observed 1,2, and 3-point change in VS, MCS-, and MCS+. I do favor including consistent movement to command to index eMCS and I appreciate the authors adding the probabilistic model in table 4. The authors have also made the code publicly available to calculate pMCID to reproduce the results. This proposed approach doesn't overestimate the clinically meaningful change.

I have the following minor suggestions/comments:

#Minor: Shouldn't MCID refer to a minimal clinically important difference? and not significant difference (in some parts of the manuscript).

#The changes in CRS-R, especially in the acute phase after brain injury, may in part be related to the changes of sedation, intracranial pressure, hemodynamics, and seizures among others. I would encourage the authors to address this point specifically. I understand that the CRS-R is not as validated to be used in the acute phase as it is for the subacute and chronic phases after injury. So perhaps the use of pMCID should be limited to those without the confounders mentioned above.

#The approach in its current version does not address the clinically meaningful change to recovered patients (although might introduce some biases), their families, clinicians, and others. The authors have addressed this point in their discussion.

#Additionally, the current approach doesn't take into account the covert responses observed with neuroimaging or electrophysiology (CMD), or covert tracking using the video eye-tracking devices (PMID: 37076304) - I also suggest adding the latter since the eye-tracking response is one of the most important elements of CRS-R. Future studies should incorporate the covert responses to build up on the proposed framework.

#I would suggest adding a paragraph on the discussion on how the authors visualize the use of this tool in future clinical studies and trials, and perhaps give an example. This will help visualize the importance of this tool.

Reviewers' comments:

Reviewer's Responses to Questions

**Comments to the Author**

1. Is the manuscript technically sound, and do the data support the conclusions?

Reviewer #1: Yes

Reviewer #2: Yes

2. Has the statistical analysis been performed appropriately and rigorously? 

Reviewer #1: Yes

Reviewer #2: Yes

3. Have the authors made all data underlying the findings in their manuscript fully available?

Reviewer #1: Yes

Reviewer #2: Yes

4. Is the manuscript presented in an intelligible fashion and written in standard English?

Reviewer #1: Yes

Reviewer #2: Yes

5. Review Comments to the Author

Reviewer #1: Monti et al. present a paper in which they develop a probabilistic score of the minimal clinically important difference (MCID) after a therapeutic intervention in patients with disorders of consciousness (DoC). They elegantly introduce the intrinsic challenges of identifying this threshold (i.e., DoC patients are either unaware or low-responsive, which means they are unable to provide subjective reports of perceived benefit), and they continue to solve this problem in a most convincing way by benchmarking their probabilistic MCID score against the clinical gold standard scale of DoC, the Consciousness Recovery Scale revised (CRS-R).

The paper is timely, important, and well-written. I am convinced that it will become an invaluable and well-cited resource for therapeutic DoC trials in the future.

I have only some very minor suggestions:

1. According to the CRS-R, visual fixation is enough to diagnose MCS- but other sources have argued that visual fixation might be a reflexive behavior and that visual pursuit is required for an MCS- diagnosis. What is the authors’ opinion?

2. Being a reviewer from Europe, I am used to the term “VS/UWS” instead of “VS” given the latter’s negative connotations. “VS” is the established term in the US, but for the sake of international outreach, the authors might want to consider changing the terminology. (I know, I know… “VS/UWS” sounds awkward, but still.)

3. The differentiation between VS/UWS and MCS is most probably gradual (continuous) rather than binary (all-or none), and the level of arousal and responsiveness in DoC patients can be subject to fluctuations over very short time spans (minutes, hours), which the MCID score for obvious reasons cannot take into account. Perhaps the authors might want to include a statement about this limitation.

4. Conclusion, line 307: “This is the first study reporting…” – I am not sure I would call this a “study” since the authors did not collect original data or tested their score in a patient cohort. I would suggest changing the sentence to “We report…” or “This is the first paper reporting…”

5. Lines 129/1300: Small typo, should be ”In what follows, we..”

6. The authors provide a link to OSF with their code to calculate the MCID score, which is excellent. A graphical depiction of the idea/concept/design of the MCID score and its calculation would be great. It's optional, but the visual appeal of the paper could be increased a lot with a nice illustration.

Reviewer #2: I read Dr. Monti et al manuscript with interest. This is an extremely important topic, and I am grateful for the authors addressing and proposing their approach of pMCID in DOC patients. I congratulate them on this very well written paper.

The two challenges as explained elegantly by the authors in the MCID concept in DOC patients are: 1/ the subjective perception of improvement by the patient is absent or not accessible, 2/ the CRS-R is not designed to be proportional and/or consistent. Thus, the authors have developed a probabilistic approach to establishing MCID (pMCID) for the CRS-R. The authors selected 8 threshold behaviors that reflect the behavioral diagnostic of each DOC (VS, MCS-, and MCS+). Tables 3 &4 reflect the probability of patient attaining a new threshold behavior given an observed 1,2, and 3-point change in VS, MCS-, and MCS+. I do favor including consistent movement to command to index eMCS and I appreciate the authors adding the probabilistic model in table 4. The authors have also made the code publicly available to calculate pMCID to reproduce the results. This proposed approach doesn't overestimate the clinically meaningful change.

I have the following minor suggestions/comments:

#Minor: Shouldn't MCID refer to a minimal clinically important difference? and not significant difference (in some parts of the manuscript).

#The changes in CRS-R, especially in the acute phase after brain injury, may in part be related to the changes of sedation, intracranial pressure, hemodynamics, and seizures among others. I would encourage the authors to address this point specifically. I understand that the CRS-R is not as validated to be used in the acute phase as it is for the subacute and chronic phases after injury. So perhaps the use of pMCID should be limited to those without the confounders mentioned above.

#The approach in its current version does not address the clinically meaningful change to recovered patients (although might introduce some biases), their families, clinicians, and others. The authors have addressed this point in their discussion.

#Additionally, the current approach doesn't take into account the covert responses observed with neuroimaging or electrophysiology (CMD), or covert tracking using the video eye-tracking devices (PMID: 37076304) - I also suggest adding the latter since the eye-tracking response is one of the most important elements of CRS-R. Future studies should incorporate the covert responses to build up on the proposed framework.

#I would suggest adding a paragraph on the discussion on how the authors visualize the use of this tool in future clinical studies and trials, and perhaps give an example. This will help visualize the importance of this tool.

6. PLOS authors have the option to publish the peer review history of their article (what does this mean?). If published, this will include your full peer review and any attached files.

Reviewer #1: **Yes: **Daniel Kondziella

Reviewer #2: **Yes: **Ayham Alkhachroum

---

## [Author Response · Author response to Decision Letter 0]

19 Jul 2023

Please see "Response to Reviewer" document for detailed response to the provided feedback.

---

## [Editor Report · Decision Letter 1]

4 Aug 2023

What is a minimal clinically important difference for clinical trials in patients with Disorders of Consciousness? A novel probabilistic approach

PONE-D-23-10843R1

Dear Dr. Monti,

We’re pleased to inform you that your manuscript has been judged scientifically suitable for publication and will be formally accepted for publication once it meets all outstanding technical requirements.

Kind regards,

Soojin Park, M.D.

Academic Editor

PLOS ONE
---

## [Editor Report · Acceptance letter]

15 Aug 2023

PONE-D-23-10843R1 

What is a minimal clinically important difference for clinical trials in patients with Disorders of Consciousness? A novel probabilistic approach 

Dear Dr. Monti:

I'm pleased to inform you that your manuscript has been deemed suitable for publication in PLOS ONE. Congratulations! Your manuscript is now with our production department. 

Kind regards, 

on behalf of

Dr. Soojin Park 

Academic Editor

PLOS ONE